# Appendiceal Intussusception: A Rare Diagnosis and the Role of Imaging in Its Detection

**DOI:** 10.3390/diagnostics15030256

**Published:** 2025-01-23

**Authors:** Stefan Milošević, Miljana Bubanja, Anja Zugic, Milica Mitrovic, Katarina Stosic, Vasko Tosic, Dragan Vasin, Aleksandra Djuric-Stefanovic

**Affiliations:** 1Center for Radiology and Magnetic Resonance Imaging, University Clinical Centre of Serbia, Pasterova No. 2, 11000 Belgrade, Serbia; bubanjamiljana97@gmail.com (M.B.); drmilica@yahoo.com (M.M.); katestosic@gmail.com (K.S.); draganvasin@gmail.com (D.V.); aleksandra.djuricstefanovic@gmail.com (A.D.-S.); 2Clinic for Digestive Surgery, University Clinical Centre of Serbia, Koste Todorovica Street, No. 6, 11000 Belgrade, Serbia; anjazugic@gmail.com; 3Department for Radiology, Faculty of Medicine, University of Belgrade, Dr Subotica No. 8, 11000 Belgrade, Serbia; 4Clinic for Emergency Surgery, University Clinical Center of Serbia, Koste Todorovica No. 6, 11000 Belgrade, Serbia; vaskotosic@gmail.com

**Keywords:** appendix, intussusception, endometriosis, diagnostic imaging

## Abstract

Appendiceal intussusception is a rare condition characterized by the invagination of the appendix into the base of the cecum. In some cases, this condition can lead to obstruction, ischemia, and perforation. It is more common in elderly patients, particularly women, and is often associated with the presence of a lesion, benign or malignant, acting as a “lead point.” This case report details the emergency management of a 54-year-old female patient with severe abdominal pain, nausea, and vomiting. The physical examination was unremarkable, as were the laboratory tests. However, ultrasound revealed a small amount of fluid in the pelvis, leading to further investigation with a CT scan which showed appendiceal intussusception with significant wall edema, fluid in the surrounding fatty tissue, and reactive lymph nodes. The patient then underwent an operation that confirmed an intussuscepted appendix showing signs of ischemia. Histopathology showed the presence of an endometrioma acting as the “lead point”. This case report showcases the crucial role of diagnostic imaging, which is superior to clinical examination and laboratory tests when diagnosing these patients. Diagnostic imaging, coupled with careful differential diagnosis, is vital to distinguishing benign conditions from malignancy and ensuring timely and appropriate intervention. Early diagnosis and surgical intervention are essential to prevent life-threatening complications such as gangrene and perforation and exclude malignancy in adult patients.

**Figure 1 diagnostics-15-00256-f001:**
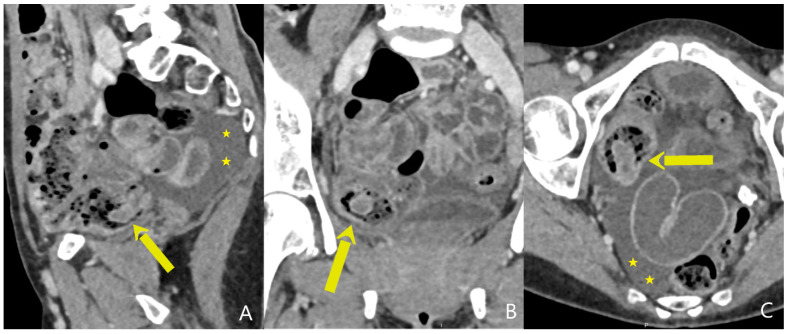
Sagittal (**A**), coronal (**B**), and axial (**C**) cross-section of the CT depicting intussusception of the whole appendix into the lumen of the cecum (arrow) with the edematous hyperattenuating wall indicating inflammation. Indirect signs of appendiceal inflammation included a moderate amount of clear free fluid in the pelvis (stars). A 54-year-old female patient presented to the Emergency Department with severe abdominal pain, predominantly localized in the upper abdomen, accompanied by nausea and vomiting. In her medical history, she stated multiple abdominal surgeries, including a surgery for endometriosis and a hysterectomy due to cervical cancer. The physical examination revealed sharp periumbilical pain radiating into the right lower quadrant, so further diagnostic tests were requested. The blood count showed mild leukocytosis (WBC: 11.3 × 10^9^/L), a normal erythrocyte count (RBC: 5.13 × 10^12^/L), and a hemoglobin level of 147 g/L. The C-reactive protein (CRP) level was within normal limits. The plain abdominal X-ray was normal, while the ultrasound examination revealed a moderate amount of clear free fluid in the pelvis, as well as edema of the antropyloric segment of the stomach. These diagnostic tests did not clarify the underlying cause of the patient’s severe abdominal pain, so further diagnostics, including a computed tomography (CT) examination of the abdomen, were requested. The abdominal CT scan, performed in three phases (non-contrast, arterial, and venous), revealed a mediopositioned cecum with conglomerated loops of small intestines around the ileocecal valve, suggestive of adhesions but with no signs of ileus. Additionally, a tubular structure starting from the base of the cecum and extending intraluminally was observed (Figure 1). This intraluminal structure had edematous, thickened walls that were hyperattenuating, suggestive of inflammation. Also, a moderate amount of clear free fluid (up to 12 Hounsfield units) could be seen in the pelvis. Based on the CT findings and the fact that the patient had no appendectomy in her medical history, intussusception of the appendix with consequent inflammation was suspected. Considering the CT findings and the fact that the patient’s overall condition was worsening, surgery was deemed necessary. The operation began laparoscopically, but, due to the presence of numerous adhesions and conglomerates of small-bowel loops, it was impossible to visualize and free the cecum and appendix, which were medially displaced, so the surgical team decided to switch to an open procedure. After conversion, extensive adhesiolysis and mobilization of the cecum were performed. The appendix was not visible at the junction of the three teniae, but after extensive dissection, its base was located, revealing that the appendix had indeed intussuscepted into the cecum. Macroscopically, it appeared phlegmonous, filled with fecaliths, and livid in color. The patient had a successful postoperative recovery and was discharged in good condition. The histopathological findings showed an inflamed appendix with signs of endometriosis in the form of endometrial stroma and glands located in the serosal and muscular layer. The endometrial tissue acted as a guiding point, causing the intussusception. Intussusception is generally defined as the invagination of one segment of the intestine with its mesenteric stalk (intussusceptum) into the lumen of an adjacent segment of the intestine (intussuscipiens), which is most commonly caused by abnormal peristalsis of that segment of the bowel [1]. This process can ultimately compromise blood vessels, particularly veins, leading to congestion of the bowel wall and later arterial occlusion. It can also disrupt the capillary network, which, if left untreated, may result in ischemia, necrosis, and perforation of the affected intestinal segment. Intussusception of the appendix is relatively rare, with a frequency of 0.01% based on a study involving 71,000 surgical specimens of the appendix [2]. The incidence is significantly higher in adults (76%) compared to children (24%), with women being more frequently affected among adults (72%), while men are more commonly affected in the pediatric population (58%) [3]. Due to the rarity of appendiceal intussusception, the literature is primarily limited to case reports and small case series, with the few existing reviews offering conflicting information. The pathophysiology of appendiceal intussusception relies on the presence of a lesion acting as a “lead point” for invagination [4]. These lesions, which are various and can include neoplasms, polyps, and endometriosis, initiate peristaltic waves that pull the lesion into the lumen of the intestine, leading to invagination [1]. In adults, the most common causes are malignant diseases, accounting for approximately 65% of cases, while inflammatory processes such as endometriosis, like in our case, and mucocele are also significant factors [5,6]. The clinical presentation of appendiceal intussusception varies from acute symptoms that mimic acute appendicitis to chronic and long-term presentations with intermittent symptoms. The majority of patients (63%) have a chronic presentation, with symptoms decreasing and reappearing over a period of several weeks to months. The most common symptoms include abdominal pain (78%), vomiting (26%), and blood in the stool (23%) [3]. Diagnosing appendiceal intussusception is challenging. Historically, most cases were identified during surgery or through pathological examination afterward. However, advancements in diagnostic imaging have enabled a shift toward preoperative diagnosis. In recent years, 58% of cases have been diagnosed preoperatively, compared to only 20% before the adoption of CT and ultrasound [4]. Double-contrast barium enema was the first diagnostic tool used to detect appendiceal intussusception. The “coil spring” sign, where the appendix represents a defect in the filling of the cecum lumen, is highly specific for this condition [7]. Although less commonly used today, barium enema still provides important information for diagnosis. Ultrasound plays a crucial role in diagnosing appendiceal intussusception, especially in pediatric patients. The characteristic “doughnut” or “target” sign on ultrasound is almost pathognomonic for intussusception [8]. Additionally, ultrasound allows dynamic monitoring of the condition and can help differentiate between intussusception and other causes of abdominal pain. CT provides high-resolution and detailed visualization of abdominal structures, essential for confirming the diagnosis of appendiceal intussusception. CT is particularly useful in adult patients where the differential diagnosis may include various malignant and benign pathologies [9,10]. CT and ultrasound often serve as complementary methods. If appendiceal intussusception is suspected on ultrasound, CT is used to confirm it. Also, potential ultrasound limitation such as artifacts from bowel gas and inter-user variability can be avoided on CT. After CT confirmation, further monitoring of the patient’s condition with ultrasound is possible in order to assess the need for potential surgery. Endoscopy offers an alternative to radiological diagnosis by allowing direct visual examination of the intestinal tract. In appendiceal intussusception, endoscopy may show the appendix as a polypoid lesion with central indentation in the cecum, which can be misleading and lead to misdiagnosis [11,12]. Therefore, the careful interpretation of endoscopic findings is necessary to avoid unnecessary or invasive interventions. Various classifications for the pathological appearance of the appendix have been suggested over time [13]. The classification is anatomical, based on which part of the appendix forms the intussusceptum and the location of the intussuscipiens. It includes five types of appendiceal intussusception (Appendix A). In our case, we could see intussusception of the whole appendix into the cecum, corresponding to type V of the anatomical classification. Although this discussion was focused on the diagnostic aspects of appendiceal intussusception, it is important to note that treatment is most often surgical, especially in cases where malignancy is suspected. Surgical resection is the most common procedure, while less-invasive methods such as barium enema or colonoscopy may be used for reducing intussusception in certain cases [14]. If a biopsy or polypectomy is performed during the endoscopy, there is a significant risk of perforation and the subsequent development of peritonitis [15], making a surgical approach the preferred option. In conclusion, recognizing and correctly diagnosing appendiceal intussusception are crucial for avoiding complications and achieving optimal patient management. Different imaging modalities provide key information but require careful interpretation to ensure accurate diagnoses and avoid unnecessary interventions. This case of appendiceal intussusception highlights the critical role of advanced diagnostic imaging and thorough differential diagnosis in facilitating early detection and management, thereby improving outcomes and guiding treatment strategies for similar rare conditions.

## Data Availability

The datasets used and analyzed in this paper are available from the corresponding author upon reasonable request.

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
