# Peer review of "Appendiceal Intussusception: A Rare Diagnosis and the Role of Imaging in Its Detection"

_diagnostics, 2025, doi:10.3390/diagnostics15030256_

Round 1

Reviewer 1 Report

Comments and Suggestions for Authors

Thank you for this eminent case presentation. I would like to congratulate with authors. Therefore, several minor revisions could provide a paper enhancement.

Abstract: The importance of differential diagnosis and imaging could be highlighted further, as they are central themes.

Introduction:

- Expand on the epidemiological aspects and challenges in diagnosis to establish a stronger background

-Add a brief mention of existing literature gaps addressed by this case.

Case Presentation:

- Include additional details about the patient’s prior surgeries (e.g., indications, complications) if relevant to the pathophysiology.

- Clarify whether alternative diagnoses were considered and ruled out during the initial evaluation.

- Integrate comparison with other diagnostic images, if applicable, to enhance discussion on imaging modalities.

Discission:

- Elaborate on potential pitfalls in imaging interpretation.

- Reflect on how this case contributes to the broader understanding or management of similar conditions.

Comments on the Quality of English Language

This paper could be accepted after major revisions according to our suggestions.

Author Response

Dear,

We would like to express our sincere appreciation for your valuable comments on our article “Appendiceal Intussusception: A Rare Diagnosis and the Role of Imaging in Its Detection”. We have given the comments serious consideration and altered the manuscript according to the suggestions.

We hope that the revised manuscript will meet your expectations and we are willing to consider all the further revisions. The revisions have been approved by all authors and the revised manuscript is attached. Thank you for your interest in our manuscript!   

Comments and answers:

Comment 1 - Abstract: The importance of differential diagnosis and imaging could be highlighted further, as they are central themes.

Answer: We added a sentence emphasizing the importance of diagnostic imaging and differential diagnosis in the abstract - line 31.

Comment 2 - Expand on the epidemiological aspects and challenges in diagnosis to establish a stronger background

Answer: From line 94-98 we have added paragraph on epidemiological aspects and challenges in diagnosis

Comment 3 - Add a brief mention of existing literature gaps addressed by this case

Answer: In line 82 we added a sentence addressing literature gaps in appendiceal intussusception.

Comment 4 - Include additional details about the patient’s prior surgeries (e.g., indications, complications) if relevant to the pathophysiology.

Answer: The patient was admitted to the emergency center as an emergency and did not have any medical documentation with her, so all data on previous surgeries were obtained through anamnesis, which is why the information is very scarce. Since the patient was operated on in another institution, precise surgical data were not available to us, but according to the patient, no complications occurred during the procedures.

Comment 5 - Clarify whether alternative diagnoses were considered and ruled out during the initial evaluation.

Answer: Due to the patients previous medical history (surgery for endometriosis and histerectomy for cervical cancer) and the fact that CT did not reveal any patological entity except intussucepted structure we did not include any other diagnosis except appendiceal intususception whit we stated in line 59.

Comment 6 - Integrate comparison with other diagnostic images, if applicable, to enhance discussion on imaging modalities.

Answer: We have added a paragraph from line 108-112 in which we compared different diagnostic modalitys and their posible integration in further patient monitoring.

Comment 7 - Elaborate on potential pitfalls in imaging interpretation.

Answer: In line 110 we added a sentence addresing potential pitfalls and limitations in imaging interpretation.

Comment 8 -Reflect on how this case contributes to the broader understanding or management of similar conditions.

Answer: In line 131 we have added a sentence explaining how this case contributed to the broader understanding or management of similar conditions.

Reviewer 2 Report

Comments and Suggestions for Authors

The present study is an interesting presentation of a rare occurence in surgical pathology.

The diagnosis is well sustained and the imaging is well explained and conducted. Also the operation is well described. 

The rest of the presentation has a good structure with differential diagnosis and treatment options taken into discussion.

Some intraoperative pictures would be a good addition for the article quality. 

Author Response

Dear,

We would like to express our sincere appreciation for your valuable comments on our article “Appendiceal Intussusception: A Rare Diagnosis and the Role of Imaging in Its Detection”. We have given the comments serious consideration and altered the manuscript according to the suggestions.

We hope that the revised manuscript will meet your expectations and we are willing to consider all the further revisions. The revisions have been approved by all authors and the revised manuscript is attached. Thank you for your interest in our manuscript!

Comment: The present study is an interesting presentation of a rare occurence in surgical pathology.

The diagnosis is well sustained and the imaging is well explained and conducted. Also the operation is well described. 

The rest of the presentation has a good structure with differential diagnosis and treatment options taken into discussion.

Some intraoperative pictures would be a good addition for the article quality. 

Answer: Thank you for the given comment. Unfortunately, intraoperative images could not be obtained because the patient was admitted late at night and there were no technical conditions to make intraoperative images.

Round 2

Reviewer 1 Report

Comments and Suggestions for Authors

Congratulations to all authors.

Comments on the Quality of English Language

It’s improved.